# Thyroid Diseases and Chronic Rhinosinusitis: A Nested Case–Control Study Using a National Health Screening Cohort

**DOI:** 10.3390/ijerph19148372

**Published:** 2022-07-08

**Authors:** Hyo Geun Choi, Tae Jun Kim, Sung Kwang Hong, Chanyang Min, Dae Myoung Yoo, Heejin Kim, Joong Seob Lee

**Affiliations:** 1Department of Otorhinolaryngology-Head & Neck Surgery, Hallym University Sacred Heart Hospital, Anyang 14068, Korea; pupen@naver.com (H.G.C.); yeramii@hanmail.net (S.K.H.); joicemin@naver.com (C.M.); 2Department of Medicine, Samsung Medical Center, Sungkyunkwan University School of Medicine, Seoul 06351, Korea; taejunk91@gmail.com; 3Hallym Data Science Laboratory, Hallym University College of Medicine, Anyang 14068, Korea; ydm1285@naver.com; 4Department of Otorhinolaryngology-Head & Neck Surgery, Dongtan Sacred Heart Hospital, Hwaseong 18450, Korea

**Keywords:** chronic rhinosinusitis, hypothyroidism, thyroid

## Abstract

This study aimed to investigate the association between thyroid diseases and chronic sinusitis (CRS) in a matched cohort in a study conducted within the Korean National Health Insurance Service-Health Screening Cohort (2002–2015). A total of 6024 patients with CRS were 1:4-matched for age, sex, household income, and region of residence with 24,096 control participants. Effects of a previous history of thyroid disease, including hypothyroidism, hyperthyroidism, thyroiditis, autoimmune thyroiditis, and Graves’ disease, were investigated using conditional logistic regression. Subgroup analyses were performed in regard to the presence of nasal polyposis. A history of hypothyroidism (2.8% vs. 1.8%), hyperthyroidism (2.0% vs. 1.5%), thyroiditis (1.1% vs. 0.8%), autoimmune thyroiditis (0.4% vs. 0.3%), and Graves’ disease (0.3% vs. 0.2%) was not more prevalent in the CRS group than in the control group according to univariate analysis (all *p* > 0.05). Hypothyroidism was associated with CRS in the multivariate-adjusted model (odds ratio [OR] 1.25, 95% confidence interval [CI] 1.00–1.57). In the subgroup analyses, thyroid diseases were not statistically significantly associated with CRS after being classified according to the presence or absence of nasal polyps. Additional studies are required to elucidate the relationship between thyroid diseases and CRS, as this may aid in screening and clinical decision making.

## 1. Introduction

It is well-known that thyroid hormones play a crucial role in energy metabolism, fetal development, and growth and influence other organs as well as nearly all parts of the human body. Therefore, thyroid hormone imbalance can cause diverse problems in the human body. Hypothyroidism, which is characterized by a deficiency in thyroid hormone levels, is a relatively common condition with a prevalence of approximately 1–3% in Western populations [1]. Most cases of hypothyroidism are caused by autoimmune thyroiditis. Secondary hypothyroidism can also be induced by iodine deficiency.

In contrast, hyperthyroidism is a condition characterized by excess thyroid hormone levels, and the prevalence of hyperthyroidism is estimated to be approximately 0.2–1.3% in iodine-sufficient areas [2]. The common cause of this state is Graves’ disease, an autoimmune disease that accounts for 60–80% of all hyperthyroidism cases. Abnormal thyroid hormone levels in patients with autoimmune thyroiditis have been associated with respiratory disorders (asthma and allergic rhinitis) [3,4] as well as with other autoimmune disorders, such as Sjögren’s disease, thrombocytopenia, rheumatoid arthritis, diabetes mellitus, systemic lupus erythematosus, and scleroderma [5].

Chronic rhinosinusitis (CRS) is one of the most common diseases among patients visiting otorhinolaryngology clinics, with a reported prevalence of 14.2% in the US and 6.95% in Korea [6,7]. Diagnosis is performed according to the presence of chronic inflammation of the nasal mucosa and paranasal sinus lasting for more than three months [8]. Phenotypically, CRS can be divided into two different subtypes: CRS with nasal polyps (CRSwNP) and CRS without nasal polyps (CRSsNP) [9]. CRSsNP is thought to result from incompletely treated or unsolved bacterial infections, or to alternatively be caused by immunodeficiency, vasculitis, or other autoimmune conditions. In contrast, CRSwNP is considered a noninfectious disorder. Most cases have a type 2 pattern of inflammation characterized by eosinophilia and elevated levels of interleukin-4, -5, and -13 cytokines [9,10].

Several studies have reported an association between CRS and autoimmune diseases such as psoriasis, lupus, rheumatoid arthritis, multiple sclerosis, and ankylosing spondylitis. A recent population-based study revealed that CRS was statistically significantly associated with premorbid autoimmune diseases [11]. Additionally, Fallahi et al. revealed that autoimmune thyroiditis is associated with a range of autoimmune diseases, such as chronic autoimmune gastritis, vitiligo, rheumatoid arthritis, multiple sclerosis, Sjögren’s disease, and systemic lupus erythematosus [12].

CRS and thyroiditis are common diseases related to autoimmune features; however, there have been few studies on their correlation. A previous case study reported evidence of an association between Hashimoto’s thyroiditis and CRS [13], and the authors discussed the potential role of the hypoactive sympathetic status seen in hypothyroidism in nasal congestion.

We conducted this study to elucidate the association between thyroid disease and CRS in a population-based cohort, as we hypothesized that autoimmunity or abnormal thyroid function might affect the course of CRS. In addition, we aimed to investigate these associations in subgroups divided according to the presence of nasal polyps.

## 2. Materials and Methods

### 2.1. Ethics

This study was approved by Hallym University’s Institutional Review Board (IRB) Ethics Committee (2019-10-023). We note that the typical requirement for written informed consent was waived by the ethics committee due to the retrospective nature of this investigation. All analyses followed the guidelines and regulations of Hallym University’s ethics committee, and adhered to the principles outlined in the Helsinki Declaration.

### 2.2. Study Population and Participant Selection

The Korean National Health Insurance Service-Health Screening Cohort has been described in detail elsewhere [14]. In brief, CRS participants were chosen from a pool of 514,866 people who had 615,488,428 medical claims codes (n = 8560). The control group was drawn from the entire non-CRS participants cohort (n = 506,306).

We excluded participants with CRS who were diagnosed in 2002 and 2003 (designated as wash-out periods, n = 2395) in order to exclusively enroll patients with CRS who had been diagnosed with this condition for the first time. Among the eligible patients with CRS, participants who were treated according to the International Statistical Classification of Diseases and Related Health Problems, 10th revision (ICD-10) diagnostic code of C73 (malignant neoplasm of the thyroid gland) were excluded from the current study (n = 141).

Among control participants, we excluded those who had died before 2004 or had no records since 2004 (n = 1518), those who were treated according to the ICD-10 diagnostic codes of J32 (chronic sinusitis) or J33 (nasal polyps) without undergoing a head and neck computed tomography (CT) evaluation (n = 124,952), and those who were treated according to an ICD-10 diagnostic code of C73 (malignant neoplasm of thyroid gland) (n = 5052).

CRS participants were enrolled at a ratio of 1:4 with control participants in terms of age, sex, income, and region of residence. To reduce selection bias, control participants were chosen at random. The index date for each participant with CRS was set as the time of CRS diagnosis. The index date for the control participants was set to the same as the index date for the matched CRS participants. During the matching process, 350,688 of the eligible control participants were excluded from the current study.

Finally, 6024 CRS participants were 1:4-matched with 24,096 control participants. In addition, we categorized patients’ CRS (and their respective matched controls) into the following subgroups: patients with CRSwNP (n = 2901) and matched controls (n = 11,604), and patients with CRS without nasal polyps (n = 3123) and matched controls (n = 12,492, Figure 1) for the purpose of subgroup analyses.

### 2.3. Exposure (Thyroid Diseases)

The thyroid diseases evaluated in our study were as follows: hypothyroidism, hyperthyroidism, thyroiditis, autoimmune thyroiditis, and Graves’ disease.

(1)Hypothyroidism was defined by the presence of the ICD-10 diagnostic codes E02 (subclinical iodine-deficiency hypothyroidism) or E03 (other hypothyroidism). We chose participants from among eligible hypothyroid patients who had been treated for hypothyroidism ≥ 2 times.(2)Hyperthyroidism was defined according to the presence of the ICD-10 diagnostic code E05 (hyperthyroidism (thyrotoxicosis)). Among eligible patients with hyperthyroidism, we selected participants who had been treated for hyperthyroidism ≥ 2 times.(3)Thyroiditis was defined according to the presence of the ICD-10 diagnostic code E06 (thyroiditis). Among eligible patients with thyroiditis, we selected participants who had been treated for thyroiditis ≥ 2 times.(4)Autoimmune thyroiditis was defined according to the presence of the ICD-10 diagnostic code E063 (autoimmune thyroiditis). Among eligible patients with autoimmune thyroiditis, we selected participants who had been treated for autoimmune thyroiditis ≥ 2 times.(5)Graves’ disease was defined according to the presence of the ICD-10 diagnostic code E050 (thyrotoxicosis with diffuse goiter). Among eligible patients with Graves’ disease, we selected participants who had been treated with anti-thyroid drugs for ≥3 months.

### 2.4. Outcome (CRS)

Participants with CRS were included in the current study if they had been diagnosed with chronic sinusitis (ICD-10 code: J32). Among eligible patients with CRS, we chose participants who had been treated for CRS ≥ 2 times and had a head and neck CT (claim codes: HA401-HA416, HA441-HA443, HA451-HA453, HA461-HA463, or HA471-HA473). CRS was classified into the following subgroups for secondary analyses: CRSwNP and CRS without nasal polyps, depending on the patients’ treatment history in regard to nasal polyps (ICD-10: J33).

### 2.5. Covariates

The age groups were divided into five-year intervals: 40–44, 45–49, 50–54, …, and 85+ years (for a total of 10 age groups). Income levels were divided into five subgroups (class 1 (lowest income) to class 5 (highest income)). Following the methodology used in our previous study [15], region of residence was classified as urban or rural. Smoking, alcohol consumption, and obesity (as measured by body mass index, BMI, in kg/m^2^) were classified similarly to our previous study [16].

Standard clinical procedures were used to test total cholesterol, fasting blood glucose levels, systolic blood pressure (SBP), and diastolic blood pressure (DBP). The levothyroxine medication prescription dates were abstracted between 2002 and 2015.

We also employed the Charlson comorbidity index (CCI), which has been widely used to measure disease burden, using information on 17 comorbidities in order to construct the continuous variable (0 (no comorbidities) through 29 (multiple comorbidities)) [17].

### 2.6. Statistical Analyses

General medical and demographic characteristics between the CRS and control groups were compared using standardized differences.

Conditional logistic regression was used to estimate odds ratios (ORs) and 95% confidence intervals (CIs) for each thyroid disease relative to total CRS, CRSwNP, and CRS without nasal polyps. Findings were calculated for a crude model, Model 1 (adjusted for total cholesterol, SBP, DBP, fasting blood glucose levels, obesity, smoking status, alcohol consumption, levothyroxine medication prescription dates, and CCI scores), Model 2 (Model 1 plus hypothyroidism, hyperthyroidism, and thyroiditis), and Model 3 (Model 1 plus hypothyroidism, autoimmune thyroiditis, and Graves’ disease). The analyses were stratified according to age, sex, income, and region of residence.

Two-tailed analyses were carried out, and statistical significance was defined as a two-sided *p*-value of less than 0.05. All statistical analyses were carried out using SAS statistical software (v. 9.4; SAS Institute Inc., Cary, NC, USA).

## 3. Results

There were no statistically significant differences in the rates of hypothyroidism, hyperthyroidism, thyroiditis, autoimmune thyroiditis, or Graves’ disease between the CRS and control groups in the current study (standardized differences: 0.06 for hypothyroidism, 0.04 for hyperthyroidism, 0.03 for thyroiditis, 0.01 for autoimmune thyroiditis, and 0.02 for Graves’ disease; Table 1). The CRS and control groups were matched in terms of age, sex, income, and region of residence (all standardized differences = 0.00). The distributions of study covariates, namely obesity, smoking status, alcohol consumption, CCI scores, total cholesterol levels, SBP, DBP, fasting blood glucose levels, and levothyroxine medication dates, were not statistically significantly different between the CRS and control groups (Table 1).

The adjusted ORs for hypothyroidism and hyperthyroidism showed associations with CRS in Model 1 (1.31, 95% CI 1.05–1.63, *p* = 0.018, and 1.28, 95% CI 1.03–1.58, *p* = 0.024, respectively). The adjusted ORs for hypothyroidism in relation to CRS were also increased in Models 2 and 3 (1.25, 95% CI 1.00–1.57, *p* = 0.050, and 1.31, 95% CI 1.05–1.64, *p* = 0.017; Table 2).

We found that a history of hypothyroidism was related to an increased OR for CRSwNP in the crude model (95% CI 1.06–1.92, *p* = 0.019). The adjusted OR of each thyroid disease in relation to CRSwNP did not statistically significantly increase in Models 1, 2, or 3 (Table 3).

In patients with CRS presenting without nasal polyps, the adjusted ORs for hypothyroidism, hyperthyroidism, thyroiditis, and autoimmune thyroiditis were increased in the crude analysis (*p* < 0.001 for hypothyroidism, *p* = 0.002 for hyperthyroidism, *p* = 0.003 for thyroiditis, and *p* = 0.038 for autoimmune thyroiditis; Table 4). In the crude model and Model 1, hyperthyroidism was associated with a higher risk of CRS (1.53, 95% CI 1.17–1.99, *p* = 0.002, and 1.38, 95% CI 0.99–1.72, *p* = 0.021; Table 4).

When adjusting for total cholesterol, SBP, DBP, fasting blood glucose levels, obesity, smoking status, alcohol consumption, levothyroxine medication prescription dates, and CCI scores, hyperthyroidism and thyroiditis were both found to be increased in patients with CRS without nasal polyps (1.38, 95% CI 1.05–1.81, *p* = 0.021, and 1.45, 95% CI 1.02–2.07, *p* = 0.041). However, the adjusted ORs for thyroid diseases in relation to CRS without nasal polyps in Models 2 and 3 were not statistically significant (Table 4).

## 4. Discussion

CRS is a common condition and is considered a multifactorial disease. A previous Taiwanese population-based study showed that CRS is associated with premorbid medical conditions such as asthma, chronic pulmonary diseases, and weight loss [18]. Among various comorbidities, hypothyroidism (OR = 1.61) was found to occur more frequently in patients with CRS. In the present nested case–control study, hypothyroidism was associated with a higher risk of CRS after adjusting for age, sex, household income, region of residence, and other comorbidities.

In the past, Chavanne et al. observed engorgement of the turbinate with pale mucosa, which was reversed with thyroxine injections in surgical thyroidectomy patients [19]. In an animal study, changes in the nasal mucosa, including hypertrophy, ciliary loss, and submucosal inflammatory cell infiltration, were observed in an iatrogenic hypothyroidism model [20]. Jeevan et al. also reported a case of a patient who was successfully treated for refractory sinusitis and was incidentally diagnosed with Hashimoto’s thyroiditis; this patient was treated with levothyroxine and hydrocortisone. The authors hypothesized that hypothyroidism’s low sympathetic status would result in a predominance of parasympathetic activity in the nose, resulting in vasodilation and nasal congestion [13].

Moreover, a recent study attempted to reveal the association between hypothyroidism and rhinitis in patients with newly developed hypothyroidism [21]. These researchers reported that hypothyroidism induced a prolonged nasal mucociliary clearance time, which is relevant to the stagnation of nasal secretions [21]. Although the exact pathophysiology of CRS is not fully understood, recurrent mucosal swelling around the sinus ostia can restrict ventilation. Moreover, a prolonged ciliary clearance time may induce mucus retention and infection.

Many studies have investigated the association between autoimmune diseases and CRS, because autoimmune diseases are thought to share the same or similar pathogenic mechanisms as those relevant to the development of CRS. In addition, autoimmune disease may contribute to a poor CRS prognosis [11]. Shih et al. previously reported that CRS was statistically significantly associated with autoimmune diseases in a Taiwanese population-based case–control study. Interestingly, in their subgroup analysis, CRS showed a positive association with autoimmune disease regardless of the presence of nasal polyps [11]. Although the possible mechanisms through which autoimmune diseases may affect the development of CRS are largely uncertain, it is plausible that underlying autoimmune diseases might affect CRS pathogenesis by altering levels of inflammatory cytokines as well as through signaling pathways [22,23]. However, in our study, we did not detect a statistically significant association between CRS and autoimmune thyroiditis (Table 2, Table 3 and Table 4). This result, which was opposite to the findings reported in previous investigations, might be attributed to the modest size of our evaluated study population with autoimmune thyroiditis. In addition, differential findings might be caused by the misclassification of autoimmune thyroiditis. For example, Hashimoto’s disease, which is classified as hypothyroidism and is not included in the definition of autoimmune thyroiditis, might be associated with some degree of misclassification.

Although CRSwNP and CRSsNP have distinct biological characteristics, we found that the presence of nasal polyps was not affected in the patients with thyroid disease enrolled in our study. CRSwNP has been reported to demonstrate features of autoimmune disorders. For example, Kato et al. found autoantibodies in patients with CRSwNP that were reactive against nuclear antigens (anti-dsDNAs) [24]. These antibodies are known to play a central role in the pathogenesis of various autoimmune disorders. In addition, Tan et al. reported microarray results showing an autoantibody response against thyroid antigens in nasal polyps [25]. In the present study, no statistically significant association was observed between thyroid disease and CRSwNP. Additional studies on this topic may help elucidate the relationship between thyroid disease and CRSwNP.

The present study has several limitations. First, exact serum levels of thyroid hormones could not be evaluated with a thyroid function test, because we used health claim data. Although many medical claims codes reflect thyroid disease diagnoses, information on the exact levels of thyroid hormones would ensure a substantially more accurate analysis. Therefore, disease severity could not be evaluated. Because information regarding serum levels of thyroid hormones ensures an exact analysis, studies incorporating information on serum thyroid hormones are recommended in the future. In addition, there can be misclassification problems, such as subclinical or untreated thyroid diseases. Second, CRS endotypes were not considered in the present study. A recent study demonstrated that CRS can be classified into five subgroups according to the presence and levels of inflammatory cytokines [26]. These authors suggested that CRS is not a dichotomous disease but is rather a multidimensional disease driven by different inflammatory mechanisms. Future studies considering the immunologic endotypes of CRS may be helpful in elucidating the mechanisms underlying the observed associations between CRS and thyroid disorders.

In addition to the limitations specified above, this study also has several notable strengths. First, this investigation was conducted within a large and representative national cohort. In population-based studies, the number of participants (i.e., statistical power) is clearly associated with the reliability of study findings. Until now, there have been only a few studies aiming to elucidate the relationship between thyroid diseases and CRS at a population level. Second, potential confounding factors were well-controlled in the current study. The participants were matched according to age, sex, income, and region of residence, and we likewise adjusted for a range of other potential confounding factors, which included not only CRS risk factors but also different thyroid conditions that might be mutually associated.

## 5. Conclusions

Our study reported a statistically significant association between hypothyroidism and CRS. However, in the subgroup analyses, thyroid disease was not statistically significantly associated with CRS categorized according to the presence or absence of nasal polyps. Future studies are required to fully elucidate the relationship between thyroid diseases and CRS, including the mechanisms underlying this association.

## Figures and Tables

**Figure 1 ijerph-19-08372-f001:**
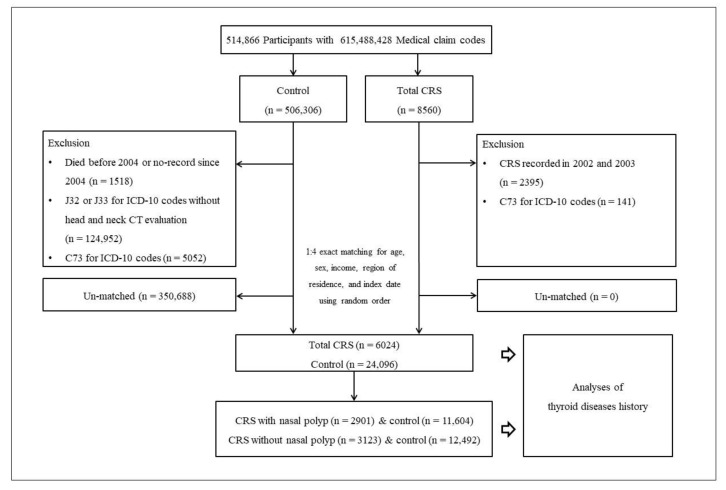
A schematic representation of the current study’s participant selection process. A total of 6024 eligible patients with chronic sinusitis (CRS) were matched with 24,096 control participants for age, sex, income, and region of residence out of a total of 514,866 participants. Study participants were regrouped into the following subgroups: patients with CRS with nasal polyps (n = 2901) and matched controls (n = 11,604), and patients with CRS without nasal polyps (n = 3123) and matched controls (n = 12,492).

**Table 1 ijerph-19-08372-t001:** General characteristics of participants.

Characteristics	Total Participants		
	Total CRS	Control	** Standardized Difference
Age (years old, n, %)			0.00
40–44	229 (3.8)	916 (3.8)	
45–49	954 (15.8)	3816 (15.8)	
50–54	1294 (21.5)	5176 (21.5)	
55–59	1273 (21.1)	5092 (21.1)	
60–64	973 (16.2)	3892 (16.2)	
65–69	694 (11.5)	2776 (11.5)	
70–74	379 (6.3)	1516 (6.3)	
75–79	167 (2.8)	668 (2.8)	
80–84	50 (0.8)	200 (0.8)	
85+	11 (0.2)	44 (0.2)	
Sex (n, %)			0.00
Males	3747 (62.2)	14,988 (62.2)	
Females	2277 (37.8)	9108 (37.8)	
Income (n, %)			0.00
1 (lowest)	741 (12.3)	2964 (12.3)	
2	711 (11.8)	2844 (11.8)	
3	918 (15.2)	3672 (15.2)	
4	1317 (21.9)	5268 (21.9)	
5 (highest)	2337 (38.8)	9348 (38.8)	
Region of residence (n, %)			0.00
Urban	2791 (46.3)	11,164 (46.3)	
Rural	3233 (53.7)	12,932 (53.7)	
Total cholesterol (mg/dL, mean, SD)	197.3 (37.3)	199.1 (37.6)	0.05
SBP (mmHg, mean, SD)	125.5 (16.1)	126.5 (16.6)	0.06
DBP (mmHg, mean, SD)	78.4 (10.6)	78.9 (10.9)	0.05
Fasting blood glucose (mg/dL, mean, SD)	99.6 (32.0)	100.6 (30.4)	0.03
Levothyroxine medication dates (days, mean, SD)	70.3 (464.7)	42.3 (349.0)	0.07
Obesity (n, %) *			0.07
Underweight	103 (1.7)	541 (2.3)	
Normal	1947 (32.3)	8386 (34.8)	
Overweight	1786 (29.7)	6728 (27.9)	
Obese I	2018 (33.5)	7785 (32.3)	
Obese II	170 (2.8)	656 (2.7)	
Smoking status (n, %)			0.07
Nonsmoker	3931 (65.3)	15,506 (64.4)	
Past smoker	896 (14.9)	3188 (13.2)	
Current smoker	1197 (19.9)	5402 (22.4)	
Alcohol consumption (n, %)			0.01
<1 time a week	3923 (65.1)	15,537 (64.5)	
≥1 time a week	2101 (34.9)	8559 (35.5)	
CCI score (score, n, %)			0.16
0	3967 (65.9)	17,590 (73.0)	
1	1002 (16.6)	3077 (12.8)	
≥2	1055 (17.5)	3429 (14.2)	
Thyroid diseases (n, %)			
Hypothyroidism	166 (2.8)	432 (1.8)	0.06
Hyperthyroidism	121 (2.0)	353 (1.5)	0.04
Thyroiditis	64 (1.1)	182 (0.8)	0.03
Autoimmune thyroiditis	23 (0.4)	77 (0.3)	0.01
Graves’ disease	16 (0.3)	43 (0.2)	0.02

CCI, Charlson comorbidity index; DBP, diastolic blood pressure; SBP, systolic blood pressure; SD, standard deviation. * Obesity (BMI, body mass index, kg/m^2^) was categorized as <18.5 (underweight), ≥18.5 to <23 (normal), ≥23 to <25 (overweight), ≥25 to <30 (obese I), and ≥30 (obese II). ** Standardized difference = difference in means or proportions divided by standard error; imbalance defined as absolute value greater than 0.20 (small effect size).

**Table 2 ijerph-19-08372-t002:** Odds ratios (95% confidence interval) of each thyroid disease for total chronic rhinosinusitis.

Characteristics	CRS	Control	Odds Ratios for CRS							
	(Exposure/Total, %)	(Exposure/Total, %)	Crude ^†^	*p*-Value	Model 1 ^†,‡^	*p*-Value	Model 2 ^†,§^	*p*-Value	Model 3 ^†,‖^	*p*-Value
Hypo- thyroidism	166/6024 (2.8)	432/24,096 (1.8)	1.56 (1.30–1.88)	<0.001 *	1.31 (1.05–1.63)	0.018 *	1.25 (1.00–1.57)	0.050 *	1.31 (1.05–1.64)	0.017 *
Hyper- thyroidism	121/6024 (2.0)	353/24,096 (1.5)	1.38 (1.12–1.70)	0.003 *	1.28 (1.03–1.58)	0.024 *	1.22 (0.99–1.52)	0.068	N/A	
Thyroiditis	64/6024 (1.1)	182/24,096 (0.8)	1.42 (1.06–1.89)	0.018 *	1.23 (0.92–1.66)	0.167	1.14 (0.84–1.54)	0.405	N/A	
Autoimmune thyroiditis	23/6024 (0.4)	77/24,096 (0.3)	1.20 (0.75–1.91)	0.452	0.98 (0.61–1.58)	0.928	N/A		0.89 (0.55–1.45)	0.641
Graves’ disease	16/6024 (0.3)	43/24,096 (0.2)	1.49 (0.84–2.65)	0.174	1.42 (0.80–2.53)	0.234	N/A		1.41 (0.79–2.52)	0.245

CCI, Charlson comorbidity index; CRS, chronic rhinosinusitis; DBP, diastolic blood pressure; SBP, systolic blood pressure. * Conditional logistic regression, significance at *p* < 0.05. † Models were stratified by age, sex, income, and region of residence. ‡ Model 1 was adjusted for total cholesterol, SBP, DBP, fasting blood glucose, obesity, smoking, alcohol consumption, levothyroxine medication prescription dates, and CCI scores. § Model 2 was adjusted for Model 1 plus hypothyroidism, hyperthyroidism, and thyroiditis. ‖ Model 3 was adjusted for Model 1 plus hypothyroidism, autoimmune thyroiditis, and Graves’ disease.

**Table 3 ijerph-19-08372-t003:** Odds ratios (95% confidence interval) of each thyroid disease for total chronic rhinosinusitis with nasal polyp.

Characteristics	CRS with Nasal Polyp	Control	Odds Ratios for CRS with Nasal Polyp							
	(Exposure/Total, %)	(Exposure/Total, %)	Crude ^†^	*p*-Value	Model 1 ^†,‡^	*p*-Value	Model 2 ^†,§^	*p*-Value	Model 3 ^†,‖^	*p*-Value
Hypo- thyroidism	61/2901 (2.1)	173/11,604 (1.5)	1.43 (1.06–1.92)	0.019 *	1.25 (0.87–1.78)	0.223	1.25 (0.87–1.79)	0.228	1.29 (0.90–1.84)	0.167
Hyper- thyroidism	44/2901 (1.5)	149/11,604 (1.3)	1.19 (0.84–1.66)	0.328	1.13 (0.80–1.59)	0.489	1.12 (0.79–1.60)	0.514	N/A	
Thyroiditis	17/2901 (0.6)	70/11,604 (0.6)	0.97 (0.57–1.66)	0.914	0.88 (0.51–1.51)	0.637	0.82 (0.47–1.43)	0.477	N/A	
Autoimmune thyroiditis	5/2901 (0.2)	37/11,604 (0.3)	0.54 (0.21–1.37)	0.195	0.44 (0.17–1.14)	0.091	N/A		0.38 (0.15–1.00)	0.051
Graves’ disease	8/2901 (0.3)	19/11,604 (0.2)	1.69 (0.74–3.85)	0.216	1.66 (0.73–3.81)	0.230	N/A		1.86 (0.80–4.33)	0.153

CCI, Charlson comorbidity index; CRS, chronic rhinosinusitis; DBP, diastolic blood pressure; SBP, systolic blood pressure. * Conditional logistic regression, significance at *p* < 0.05. † Models were stratified by age, sex, income, and region of residence. ‡ Model 1 was adjusted for total cholesterol, SBP, DBP, fasting blood glucose, obesity, smoking, alcohol consumption, levothyroxine medication prescription dates, and CCI scores. § Model 2 was adjusted for Model 1 plus hypothyroidism, hyperthyroidism, and thyroiditis. ‖ Model 3 was adjusted for Model 1 plus hypothyroidism, autoimmune thyroiditis, and Graves’ disease.

**Table 4 ijerph-19-08372-t004:** Odds ratios (95% confidence interval) of each thyroid disease for total chronic rhinosinusitis without nasal polyp.

Characteristics	CRS without Nasal Polyp	Control	Odds Ratios for CRS without Nasal Polyp							
	(Exposure/Total, %)	(Exposure/Total, %)	Crude ^†^	*p*-Value	Model 1 ^†,‡^	*p*-Value	Model 2 ^†,§^	*p*-Value	Model 3 ^†,‖^	*p*-Value
Hypo- thyroidism	105/3123 (3.4)	259/12,492 (2.1)	1.66 (1.31–2.09)	<0.001 *	1.32 (1.00–1.75)	0.054	1.23 (0.92–1.64)	0.156	1.29 (0.97–1.71)	0.083
Hyper- thyroidism	77/3123 (2.5)	204/12,492 (1.6)	1.53 (1.17–1.99)	0.002 *	1.38 (1.05–1.81)	0.021 *	1.31 (0.99–1.72)	0.060	N/A	
Thyroiditis	47/3123 (1.5)	112/12,492 (0.9)	1.70 (1.20–2.40)	0.003 *	1.45 (1.02–2.07)	0.041 *	1.34 (0.93–1.93)	0.114	N/A	
Autoimmune thyroiditis	18/3123 (0.6)	40/12,492 (0.3)	1.81 (1.03–3.16)	0.038 *	1.51 (0.85–2.67)	0.161	N/A		1.39 (0.78–2.49)	0.262
Graves’ disease	8/3123 (0.3)	24/12,492 (0.2)	1.33 (0.60–2.97)	0.481	1.22 (0.54–2.74)	0.630	N/A		1.22 (0.54–2.73)	0.636

CCI, Charlson comorbidity index; CRS, chronic rhinosinusitis; DBP, diastolic blood pressure; SBP, systolic blood pressure * Conditional logistic regression, significance at *p* < 0.05. † Models were stratified by age, sex, income, and region of residence. ‡ Model 1 was adjusted for total cholesterol, SBP, DBP, fasting blood glucose, obesity, smoking, alcohol consumption, levothyroxine medication prescription dates, and CCI scores. § Model 2 was adjusted for Model 1 plus hypothyroidism, hyperthyroidism, and thyroiditis. ‖ Model 3 was adjusted for Model 1 plus hypothyroidism, autoimmune thyroiditis, and Graves’ disease.

## Data Availability

Restrictions apply to the availability of these data. Data were obtained from Korean Genome and Epidemiology Study (KoGES) and are available at (www.kdca.go.kr) (accessed on 23 May 2021), with the permission of (KoGES).

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
