# Peer review of "Thyroid Diseases and Chronic Rhinosinusitis: A Nested Case–Control Study Using a National Health Screening Cohort"

_ijerph, 2022, doi:10.3390/ijerph19148372_

Round 1

Reviewer 1 Report

Manuscript Title “Thyroid diseases and chronic rhinosinusitis: A nested case-control study using a national health screening cohort”

General comment:

This manuscript was written on the Korean population-based case-control groups of comparing. The aim is simple and clear. The thyroid disease and chronic sinusitis (CRS) in a match cohort. The specific characteristic is precise and solid to illustrate the quality. The ratio is roughly maintained 1to 4 as case to control groups.

Specific comment:

1.     Tab. 2, in the characteristics, does hyperthyroidism include those who take the thyroidectomy or not? Please specify

2.     As author stated in L301, “The present study has several limitations. First, serum levels of thyroid hormones were not evaluated. Moreover, thyroid diseases were classified using medical claims data. Although many medical claims codes reflect thyroid disease diagnoses, information on the exact levels of thyroid hormones would ensure a substantially more accurate analysis. However, our use of medical claims codes means that exact thyroid hormone levels (i.e., thyroid function test results) were not analyzed in the present study” this should be also intensified in the introduction as presumption to declare the limitation of this analysis.

3.     L316, statements here is too aggressive to imply the intention, suggested change to a smoother address, especially ”the first to elucidate” is too sensitive to address in the academic article, please revise.

Author Response

We thank the reviewer for the kind and detailed review for our manuscript. We will implement the suggestions wherever possible. 

Specific comment:

  1. Tab. 2, in the characteristics, does hyperthyroidism include those who take the thyroidectomy or not? Please specify

- Response 1:
Our data is based on Korean national health insurance service-health screening cohort. As described in materials and methods section, we excluded participants who were treated according to an ICD-10 diagnostic code of C73 (malignant neoplasm of thyroid gland).
Of course, there can be few patients who underwent thyroidectomy with other benign causes, but we think that the case would be small number, and does not impact the results.

  1. 2.As author stated in L301, “The present study has several limitations. First, serum levels of thyroid hormones were not evaluated. Moreover, thyroid diseases were classified using medical claims data. Although many medical claims codes reflect thyroid disease diagnoses, information on the exact levels of thyroid hormones would ensure a substantially more accurate analysis. However, our use of medical claims codes means that exact thyroid hormone levels (i.e., thyroid function test results) were not analyzed in the present study” this should be also intensified in the introduction as presumption to declare the limitation of this analysis.

   - Response 2:
We agree that our limitation of study, and in the introduction part, we just suggest the needs of this study and our purpose of the study. In the phrase of purpose of study, we mentioned “We conducted this study to elucidate the association between thyroid disease and CRS in a population-based cohort”. Most of population-based study uses medical claims, so patients were classified with their ICD-10 diagnostic code, not exact levels of thyroid hormones.
Instead of intensifying limitation in the introduction part, we discussed more about limitation at discussion part.
– “First, exact serum levels of thyroid hormones could not be evaluated by thyroid function test, because we used health claim data.” “In addition, there can be misclassification problems, such as subclinical or untreated thyroid diseases. “

  1. L316, statements here is too aggressive to imply the intention, suggested change to a smoother address, especially ”thefirst to elucidate” is too sensitive to address in the academic article, please revise.

   - Response 3:
We agree with your comment, and we revise the mention ”the first to elucidate” to “Until nowadays, there have been only few studies to elucidate the relationship between thyroid diseases and CRS at a population level”.

Reviewer 2 Report

hello

interesting paper, a huge number of patients

some info about primary hyperparathyroidism and occurence of Brown Tumors, espoecially in maxillary bones and co-existing polyps/sinus mucous association should be written

what alse exclusion criteria? only patients suffering from CRS for first time?

what are the study limitations?

how does income and residency status influence on endocrinal disease and nasal polyps?

why teeth influence, on dental related sinus inflammation was not mentioned?

are there any environmental or other related factors which could impact on the presented results?

how many % or RTH-therapy in head neck patients or thyroid ca can result in thyroid insuficiency and develope for example maxillary bone radio necrosis and secondary polyps? is it worth to mention this fact?

some ct/ rtg scans of an example of nasal/maxillary polyps should be uploaded, same as a picture of endoscopy with polyps and an example of what ORL/otolaryngological steps had been carried away to confirm the presence and scope/degreee of nasal polyps?

what impact of future studies this paper highlights?

are they are more addiotional data/steps, that might be improved or used in future studies to emphasize any potential endocrinal role in CRS?

overall intersting paper, but lacking some clinical relevance with other major/minor factors which might influence on CRS

good paper, needs re-arrangement

Author Response

Response to Reviewer 2 comments

interesting paper, a huge number of patients

#1. some info about primary hyperparathyroidism and occurence of Brown Tumors, espoecially in maxillary bones and co-existing polyps/sinus mucous association should be written

  • Response 1:
    Thanks for your kind comment, and I agree that hyperparathyroidism might affect the occurrence of maxillary bone and sinus problems. Unfortunately, in this time, we investigated only thyroid diseases such as hypothyroidism, hyperthyroidism, thyroiditis, autoimmune thyroiditis and Graves’ disease, not including parathyroid disease. As you mentioned, we thankfully planned to investigate the relationship of parathyroid disease and sinus problem in further next study.

#2. what else exclusion criteria? only patients suffering from CRS for first time?

  • Response 2:
    As we presented at the study population and participant selection part in methods and figure 1, we excluded 1) “participants with CRS who were diagnosed in 2002 and 2003 (designated as wash-out periods, n = 2,395) in order to exclusively enroll patients with CRS who had been diagnosed with this condition for the first time”.
    2) participants who were treated according to an ICD-10 diagnostic code of C73 (malignant neoplasm of the thyroid gland) were excluded from the current study (n = 141).

  • In control group, we excluded
    1) those who had died before 2004 or had no records since 2004 (n = 1,518),
    2) those who were treated according to the ICD-10 diagnostic codes of J32 (chronic sinusitis) or J33 (nasal polyps) without undergoing a head and neck computed tomography (CT) evaluation (n = 124,952),
  • 3) those who were treated according to an ICD-10 diagnostic code of C73 (malignant neoplasm of thyroid gland) (n = 5,052).

#3. what are the study limitations?

  • Response 3:
    We mentioned about the limitations of this study at discussion part,
    First, exact serum levels of thyroid hormones could not be evaluated by thyroid function test, because we used health claim data. Although many medical claims codes reflect thyroid disease diagnoses, information on the exact levels of thyroid hormones would ensure a substantially more accurate analysis. Therefore, disease severity could not be evaluated. Because information regarding serum levels of thyroid hormones ensures an exact analysis, studies incorporating information on serum thyroid hormones are recommended in the future. In addition, there can be misclassification problems, such as subclinical or untreated thyroid diseases.
    Second, CRS endotypes were not considered in the present study. A recent study demonstrated that CRS can be classified into five subgroups according to the presence and levels of inflammatory cytokines. These authors suggested that CRS is not a dichotomous disease but is rather a multidimensional disease driven by different inflammatory mechanisms.

#4. how does income and residency status influence on endocrinal disease and nasal polyps?

  • Response 4:
    We used the national health screening cohort study. To increase the statistical power, large study population guaranteed a sufficient number of control population matched for all possible cofounder, such as age, sex, income, and region of residence. We tried to reduce the environmental factor as a covariate.

#5. why teeth influence, on dental related sinus inflammation was not mentioned?

  • Response 5:
    I agree that dental problems can cause the sinus inflammations. The study is population-based study, so we have to investigate using the health claims. For those reasons, we cannot check every dental problem in this time. In huge cohort, we briefly wanted to reveal the association of thyroid diseases and sinusitis.

#6. are there any environmental or other related factors which could impact on the presented results?

  • Response 6:
    We agree that environmental or other possible relation factor can influence on the results. However, we used the huge cohort and we tried to adjust the covariate factor (matched for age, sex, income, and region of residence).

#7. how many % or RTH-therapy in head neck patients or thyroid ca can result in thyroid insuficiency and develope for example maxillary bone radio necrosis and secondary polyps? is it worth to mention this fact?

  • Response 7:
    We excluded patients with thyroid cancer in this study, so our participants did not performed radioiodine therapy. As you mentioned, if thyroidectomy patients got radioiodine therapy, it can affect the sinus mucosa and secondary bone lesion, and it can cause the sinus diseases. Further study would be needed for reveal the relationship between radioiodine therapy and sinus diseases.

#8. some ct/ rtg scans of an example of nasal/maxillary polyps should be uploaded, same as a picture of endoscopy with polyps and an example of what ORL/otolaryngological steps had been carried away to confirm the presence and scope/degreee of nasal polyps?

  • Response 8:
    These results were based on Korean National Health Insurance Service, and we used ICD10 diagnosis based on health claims. The diagnosis might be different from each doctor who made a diagnosis, and we did not know the severity of CRS from health claims. But we think that it cannot affect the results because the large cohort number.

#9. what impact of future studies this paper highlights?

  • Response 9:
    We think that this study have several strong points. This investigation was conducted within a large and representative national cohort. In population-based studies, the number of participants (i.e., statistical power) is clearly associated with the reliability of study findings. Until nowadays, there have been only few studies to elucidate the relationship between thyroid diseases and CRS at a population level.
    Second, potential confounding factors were well controlled in the current study. The participants were matched according to age, sex, income, and region of residence, and we likewise adjusted for a range of other potential confounding factors, which included not only CRS risk factors but also different thyroid conditions that might be mutually associated.

#10. are they are more addiotional data/steps, that might be improved or used in future studies to emphasize any potential endocrinal role in CRS?

  • Response 10:
    As we mentioned in discussion part, our data cannot know the exact serum thyroid hormone levels, so we did not check the severity of thyroid diseases. In future study, relationship between the thyroid hormone levels and sinus disease would be expected to support our results.

overall intersting paper, but lacking some clinical relevance with other major/minor factors which might influence on CRS

good paper, needs re-arrangement

  • We thank the reviewer for the kind and detailed review. We tried to implement the recommendations wherever possible, and we think that it helped to make our manuscript more comprehensive and valuable.
